# Current Trend in Antiviral Therapy for Chronic Hepatitis B

**DOI:** 10.3390/v14020434

**Published:** 2022-02-21

**Authors:** Rong-Nan Chien, Yun-Fan Liaw

**Affiliations:** Liver Research Unit, Chang Gung Memorial Hospital and University College of Medicine, 199 Tung Hwa North Road, Taipei 105, Taiwan

**Keywords:** chronic hepatitis B, combined HBsAg/ALT kinetics, finite NUC therapy, nucleos(t)ide analogue (NUC), HBsAg loss, off-NUC flare

## Abstract

Since active hepatitis B virus (HBV) replication is the key driver of hepatic necroinflammation and disease progression, the treatment aim of chronic hepatitis B (CHB) is to suppress HBV replication permanently to prevent hepatic decompensation, liver cirrhosis and/or hepatocellular carcinoma and prolong survival. Currently, pegylated interferon (Peg-IFN), entecavir (ETV), tenofovir disoproxil fumarate (TDF) and tenofovir alafenamide (TAF) are the first-line drugs of choice. Peg-IFN therapy has been used rarely due to its subcutaneous injection and significant side effect profile. Once daily oral ETV, TDF and TAF can suppress HBV DNA profoundly but have no direct action on cccDNA of the HBV-infected hepatocytes, hence continuing long-term therapy is usually needed to maintain HBV suppression, but the ultimate goal of HBsAg loss was rarely achieved (10 year 2%). In addition, long-term NUC therapy comes with several concerns such as increasing cost, medication adherence and loss-to-follow-up. Studies, mainly from Taiwan, have shown that finite NUCs therapy of two to three years in HBeAg-negative patients is feasible, safe and has a great benefit of much increasing HBsAg loss rate up to 30%/5 year. These have led an emerging paradigm shift to finite NUC therapy in HBeAg-negative patients globally. However, off-NUC relapse with hepatitis B flares may occur and have a risk of decompensation or even life-threatening outcomes. Therefore, proper monitoring, assessment, and retreatment decisions are crucial to ensure safety. Ideally, retreatment should be not too late to ensure safety and also not too early to allow further immune response for further HBsAg decline toward HBsAg loss. Assessment using combined HBsAg/ALT kinetics during hepatitis flare is better than biochemical markers alone to make a right retreatment decision. The strategy of finite NUC therapy has set a benchmark of high HBsAg loss rate to be achieved by the new anti-HBV drugs which are under preclinical or early phase study.

## 1. Introduction

Chronic infection with Hepatitis B virus (HBV) is a major global health problem, an important cause of morbidity and mortality sequelae such as hepatic decompensation, liver cirrhosis (LC) and/or hepatocellular carcinoma (HCC) development [1]. It affects ~290 million people worldwide; 64% of them reside in the Asia Pacific region, and approximately 2 billion people have been infected worldwide and about 1 million die from it annually [2]. In the past decades, basic and clinical studies have provided a better understanding of the virus, its natural history and the immunopathogenesis of chronic HBV infection [1,2,3]. Furthermore, primary prevention of HCC by universal HBV vaccination program was conducted globally and was successful in achieving declining HBV infection rates [3,4]. In addition, treatment of patients with chronic Hepatitis B (CHB) has been evolving rapidly in the past decades, with an increasing range of treatment options and the availability of multiple antiviral agents [2,3,5]. The long-term goal of antiviral therapy has partly achieved by effectively reducing the incidence of HCC development [6]; however, the existing antiviral therapy is still far from satisfactory now that new strategies and/or new drugs are emerging. This review aims to assess and discuss current and evolving trends in the antiviral therapy of chronic HBV infection.

## 2. Natural History and Treatment Considerations

Chronic HBV infection is a dynamic process of the interactions between HBV and host immune response on hepatocytes. The hepatitis activity with the rising of serum alanine aminotransferase (ALT) is reflecting the host endogenous immune ability against HBV. The classical natural course of chronic HBV infection is divided into the immune tolerance phase (high levels of HBsAg quantification [qHBsAg], seropositive for Hepatitis B e antigen [HBeAg], high levels of serum HBV DNA, normal ALT and minimal hepatic histological changes), the immune clearance phase (declining levels of qHBsAg, HBeAg seropositive, declining levels of serum HBV DNA, fluctuation of ALT and increased hepatic lobular and/or periportal activity), and followed by HBeAg seroconversion to the residual integration phase (lower levels of qHBsAg, seropositive for antibody against HBeAg [anti-HBe], lower levels of serum HBV DNA, normal ALT and minimal hepatic histological activity [7], which may reactive to HBeAg-negative CHB (reactive phase: HBeAg-negative, elevated ALT with increased hepatitis activity and higher HBV DNA level) [1]. The severity, extent, duration and frequency of hepatic lobular alterations during hepatitis flares are determinants of disease progression (LC and/or HCC), remission or clearance of HBV antigens [1].

Obviously, active HBV replication is the key driver leading to hepatic necroinflammation and disease progression. Therefore, HBeAg-positive and HBeAg-negative patients with CHB are candidates of anti-HBV therapy with an aim of permanent suppression of HBV replication. The short-term goal of treatment is profound HBV DNA suppression, ALT normalization, the prevention of hepatic decompensation and the achievement of HBeAg seroconversion in HBeAg-positive patients. The long-term goal of therapy is to prevent hepatic decompensation, reduce progression to LC and/or HCC and prolong survival [8]. The ultimate goal of antiviral therapy is HBsAg loss, considered as a functional cure, with or without seroconversion to its antibody (anti-HBs). 

## 3. Currently Available Antivirals

Currently, two subcutaneous interferon (IFN)-based drugs and seven oral nucleos(t)ide analogues (NUC) have been approved for the treatment of CHB. Lamivudine (LAM), adefovir (ADV) and telbivudine (LdT) have been rendered almost obsolete because of issues of drug resistance, whereas entecavir (ETV), tenofovir disoproxil fumarate (TDF), tenofovir alafenamide (TAF) and pegylated interferon (Peg-IFN) are the first-line drugs of choice for anti-HBV therapy [5,9,10,11].

### 3.1. IFN-Based Therapy

Peg-IFN has the benefit of both antiviral and immunomodulatory effects after a finite course of therapy in HBeAg-positive and -negative CHB patients. Studies showed that HBeAg-positive CHB patients treated with subcutaneous Peg-IFN 180 μg per week for 48 weeks achieved ALT normalization in 41%, HBeAg seroconversion with HBV DNA <400 copies/mL in 32% and HBsAg seroconversion in 3%, assessed at 24 weeks post-treatment [12]. The responses of Peg-IFN therapy at shorter duration (24 weeks) and/or lower weekly dose (90 μg) were inferior to the recommended 180 μg/week for 48 weeks regimen [13]. In addition, a mean three years interval follow-up study in those HBeAg-seropositive patients after 52 weeks Peg-IFN ± LAM treatment has revealed that 81% of those initial HBeAg loss had sustained its response and 27% of those initial HBeAg non-responders had achieved delayed HBeAg loss. Of note is that 30% of the initial responders or 11% of overall patients achieved HBsAg loss thereafter [14]. Wong et al. further reported the durability at five years post-therapy, showing 69% of the initial HBeAg non-responders followed by delayed HBeAg seroconversion with a 60% overall HBeAg seroconversion rate [15]. Concerning HBeAg-negative CHB patients, the phase III global trial have demonstrated that ALT normalization, HBV DNA <20,000 copies/mL, HBV DNA <400 copies/mL and HBsAg loss were 59%, 43%, 19% and 3%, respectively, after six months post-treatment (Peg-IFN therapy at 180 μg/week for 48 weeks) assessment [16]. Upon long-term follow-up, 31% and 23% of patients treated with 1-year Peg-IFN ± LAM achieved HBV DNA <2000 IU/mL at 1- and 5-years post-therapy, respectively, and HBsAg loss in 5% and 12% at one and five years post-therapy, respectively [17].

### 3.2. NUCs Therapy

#### 3.2.1. Entecavir

ETV, a cyclopentyl guanosine analogue, is a potent inhibition of the HBV polymerase, achieving a mean 6.9 log decline of serum HBV DNA. An international collaborative ETV treatment in HBeAg-positive patients showed a five-year cumulative probability of HBV DNA < 300 copies/mL of 94% [18]. Another report from Hong Kong involving 222 treatment-naïve patients treated with five-year ETV showed undetectable HBV DNA in 97%, achieving HBeAg seroconversion in 66.9% and HBsAg loss in one patient [19]. Other studies also demonstrated undetectable HBV DNA of 83–90%, and HBeAg seroconversion of 24–44% at year three of treatment [20,21,22]. But HBsAg loss was rare, and this was reported in only 0–1.4% of HBeAg-positive patients after three to five years of ETV therapy [20,21]. In addition, improvement of hepatic activity and fibrosis could be achieved after continuous ETV treatment [23]. However, Papatheodoris et al. had demonstrated that the calculated HBsAg decline is only 0.09 log IU/mL per year, which explains why long-term therapy is needed and the HBsAg loss rate in ETV-treated patients is low [24]. Paired histologic studies showed fibrosis regression and cirrhosis remission in 85% and 100%, respectively, of patients after 3–7 years ETV therapy [23,25]. Compared with historical untreated controls among patients with or without liver cirrhosis, several long-term cohort studies from Asia have demonstrated that ETV-treated patients could reduce the incidence of liver related complications and improve their survival [26,27,28]. Moreover, a multicenter collaborative study from Taiwan also showed a significant reduction in the risk of cirrhotic complications, HCC and mortality in patients with HBV-related cirrhosis after four-year ETV therapy [29].

#### 3.2.2. Tenofovir Disoproxil Fumarate

TDF is an acyclic adenine nucleotide analogue effective for both HBV and human immune deficiency virus (HIV). In a phase III randomized trial, TDF 300 mg daily, compared to ADV 10 mg daily, has been shown to have superior HBV DNA suppression in both HBeAg-positive and HBeAg-negative patients [30]. TDF treatment for seven years achieved undetectable HBV DNA in 99.3%, ALT normalization in 80%, HBeAg loss of 54.5% and HBsAg loss in 11.8% of patients with HBeAg loss [31]. In HBeAg-negative patients, the calculated HBsAg decline was only −0.09 log_10_ IU/mL [32] and only one (0.3%) of 375 patients was reported to achieve HBsAg loss after seven years TDF therapy [31]. More importantly, Marcellin et al. had reported that 74% of cirrhotic patients showed regression of liver cirrhosis and 87% of patients improved histological activity of liver on paired liver biopsy at five years, [33]. Based on the REACH-B risk calculator, Kim et al. showed a reduced incidence of HCC after TDF long-term therapy among patients without cirrhosis [34].

#### 3.2.3. Tenofovir Alafenamide

Recently, a small molecular weight oral prodrug of tenofovir, TAF, has been approved for the treatment of chronic HBV infection to replace long-term TDF therapy because of some concerns with regard to renal injury and decreased bone mineral density. Compared to a 300 mg dose of TDF administration, a 25 mg dose of TAF has showed more than 90% lower systemic tenofovir concentration and higher intracellular concentration during pharmacokinetic study [35]. A randomized, double-blind non-inferior trial in both HBeAg-positive and HBeAg-negative patients receiving TAF or TDF therapy have demonstrated similar rates of achieving an HBV DNA levels <29 IU/mL at week 48 [36,37]. However, using AASLD ALT normal criteria (male: ALT ≤ 30 U/L and female: ALT ≤ 19 U/L), TAF is associated with higher rate of ALT normalization than TDF (*p* < 0.05) [37]. Concerning the renal safety using the estimated glomerular filtration rate (eGFR) analysis, patients with TAF had a smaller reduction than those with TDF therapy. In comparison with decreased in bone mineral density at the hip and spine, patients with TAF also showed smaller than those with TDF therapy significantly in both HBeAg-positive and HBeAg-negative patients [36,37].

## 4. Problems of Current Therapy

### 4.1. IFN-Based Therapy

Although it is recommended as one of the first line treatments by all therapeutic guidelines, it has been limited by its subcutaneous injection, poor tolerability and significant side effect profile. In addition, it is also contraindicated in patients with hepatic decompensation, immunosuppressed states, major comorbid diseases, and pregnancy. Hence, Peg-IFN has been used in only <5% of the real world patients and is only preferred by young patients who wish to have children in the near future and those who refuse long-term NUCs treatment. 

### 4.2. NUCs Therapy

Since NUCs can suppress HBV DNA profoundly but have no direct action on cccDNA of the HBV-infected hepatocytes, the indefinite continuation of long-term NUCs therapy is usually necessary to maintain a virological response [5,9,10,11]. A mathematic modelling study has estimated that three to four decades of continuous NUC therapy would be needed to achieve a functional cure [38]. However, several concerns and disadvantages of life-long NUC therapy have emerged and been extensively discussed [39], as summarized in the following subsections.

#### 4.2.1. Ultimate Goal of HBsAg Loss Rarely Achieved

A large multinational multicenter long-term ETV/TDF therapy cohort study in 4769 patients demonstrated a 10-year HBsAg loss rate of 2.1% and an annual incidence of only 0.22% [40].

#### 4.2.2. Cost and Drug-Resistance Issues

Earlier reports of low genetic barrier NUCs therapy such as LAM, ADV or LdT, showed that resistance mutants may emerge [5], and may cause hepatitis, hepatitis flare, and even life-threatening hepatic decompensation [41]. In addition, peoples from resource limited countries or regions such as Asia could not afford the financial burden of long-term NUC therapy [42]. 

#### 4.2.3. Adherence and Other Additional Concerns

Even though ETV, TDF or TAF shows a high genetic barrier, and drug resistance is no longer a concern during monotherapy, drug adherence and compliance issues are emerging concerns to a physician [43]. Ford et al. performed a systematic review involving 30 studies with meta-analysis [44], and had showed an overall adherence of 74.6%, which is far from the optimal adherence of 95%. Another important study by Shin et al. from Korea involving 894 patients further showed patients with poor adherence (<70%) in 10.5% and moderate adherence (70–90%) in 20.5% during 5-year ETV therapy, respectively. More important is that such patients showed increased incidence of cirrhotic complications, HCC, and mortality significantly in a dose-dependent manner [45]. Furthermore, Ahn et al. also demonstrated that 7.3% of 658 patients (Asians: 83.3%) stopped therapy by themselves or were lost to follow-up during a five-year ETV study [46]. It is very clear, without off-therapy monitoring, that such patients may encounter some risk of severe ALT relapse, hepatic decompensation, even hepatic failure and mortality [47]. 

Together with the financial issue, these human nature related inevitable problems may get worse upon longer duration of therapy. In addition, the safety issues of NUC therapy beyond 10 years are largely unknown and is also one of the concerns.

## 5. Current Trend of NUC Therapy

### 5.1. Finite NUC Therapy 

All guidelines of major liver association have recommended that NUC therapy in HBeAg-positive CHB can be stopped after HBeAg loss with undetectable HBV DNA and consolidation therapy >12 months [9,10,11]. Along this line and against other guidelines, the APASL guidelines recommended to consider stopping NUC therapy in HBeAg-negative patients after treatment of at least two years with undetectable HBV DNA documented on three separate occasions each >6 months apart (total >12 months) with off-therapy ALT monitoring monthly in the first three months and then every three months along with an HBV DNA assay, the so called APASL stopping rule [8] Planned NUCs cessation in HBeAg-negative patients with maintained HBV suppression over one to three years may encounter virologic relapse (VR: HBV DNA > 2000 IU/mL), which may coincide with or be followed by clinical relapse (CR: VR + ALT > 2 × ULN) and hepatitis flare (ALT > 5 × ULN). The one-year rate of hepatitis flare was around 36% and the rate of ALT > 10 × ULN was around 21% in small studies (<100 patients) [39]. Our large study involving 691 patients (308 with cirrhosis), so far the largest on this issue, showed clinical relapse in 419 (61%), hepatitis flare (ALT > 5 × ULN) in 280 (41%), total bilirubin >2 mg/dL in 72 (10%), prolongation of prothrombin time with international normalized ratio (INR) > 1.5 in 16 (2%) during a median follow-up of three years after end-of-treatment (EOT), and the calculated annular incidence of hepatic decompensation was 0.28%, with a five-year cumulative incidence of 0% in 383 patients with CHB and 2.95% in 308 patients with cirrhosis, during a median follow-up of 155 (2–614) weeks [48]. These studies have shown that finite NUC therapy in HBeAg-negative patients is feasible and reasonably safe, even in patients with cirrhosis [39].

Of note is that earlier report by Hadziyannis et al. [49] that demonstrated ALT flares in 25 (76%) of the 33 patients after cessation of four-to-five-year ADV treatment with a 39% of HBsAg loss rate subsequently during a 5.5 year follow-up. Also noteworthy is that “no-retreatment “is the most important determinant for HBsAg loss. Furthermore, Honer zu Siederdissen C et al. [50] has reported a 20% HBsAg loss rate after discontinuation of NUC treatment in HBeAg-negative CHB patients during a median of 33 months follow-up. Recently, Berg et al. reported an important result of a randomized controlled trial (FINITE study). It showed HBsAg loss in 19% within three-years after cessation of TDF therapy in comparison to 0% in those who continued TDF treatment [51]. Although this is a small number study, it has provided a meaningful data of well controlled head-to-head comparison, and further confirms the pivotal findings of Hadziyannis et al. [49] that discontinuation of NUC therapy in HBeAg-negative patients with HBV DNA suppression may subsequently increase HBsAg loss rate. In addition, there were several reports worldwide showing higher HBsAg loss rate after finite NUC treatment [52,53,54,55], as summarized in Table 1. We also reported a large-scale study involving 691 HBeAg-negative patients and demonstrated that the incidence of HBsAg loss was highest in patients with sustained response (6-year: 36%) off-NUC treatment. It is worth to note that patients with clinical relapse but were not retreated had a 7.34 times higher incidence of HBsAg loss than those who received retreatment (six-year: 19% vs. 1%), confirming that “no-retreatment” is an important factor for HBsAg loss [48], as compared in Table 2. The strategy of finite NUC therapy in HBeAg-negative patients has been increasingly accepted by non-APASL countries since 2016 [10,11]. Clearly, a paradigm shift from indefinite long-term/life-long NUC therapy to finite therapy in HBeAg-negative patients is emerging [39]. European experts even consider that the strategy of finite therapy is not only an option but also becoming a more specific recommendation in future guidelines [56].

### 5.2. Off-Therapy Management and Re-Treatment Decision

Although finite NUC therapy in HBeAg-negative patients may minimize the concerns/problems of indefinite long-term therapy and much increased HBsAg loss, CR may occur in about 60% and hepatitis flare in 40% of HBeAg-negative patients after cessation of NUC therapy, and may sometimes deteriorate to hepatic decompensation and even death [48,55]. Hence, off-therapy monitoring is of paramount importance, and timely retreatment is required in some patients with hepatitis flare to prevent hepatic decompensation. It is therefore pertinent to follow the patients with rising ALT or ALT > 5 × ULN either weekly or biweekly with assays of ALT, bilirubin and INR to detect hepatic decompensation and start treatment/retreatment in time to ensure safety [57]. Given that “no-retreatment” is an important determinant for subsequent HBsAg loss [48,49], this monitoring plan can avoid unnecessary retreatment [58]. Therefore, the decision of retreatment is crucial for the best result of finite NUC therapy. Ideally, it should be not too late to prevent/rescue hepatic decompensation and not too early to allow further HBsAg decline toward HBsAg loss [39].

#### 5.2.1. Biochemical Markers for Retreatment Decision

In principle, the indications of retreatment are similar to the indications of treatment for treatment-naïve patients, such as viremic patients with persistent/intermittent ALT elevation >3 months [9,10,11]. However, patients with hepatitis flare are at risk of hepatic decompensation and hence require special consideration. Various biochemical criteria have been applied in studies on off-NUC retreatment [51,59,60,61], as shown in Table 3. Retreatment decisions by these criteria determined at one single time point may include patients destined to have spontaneous beneficial outcomes. In contrast, retreatment decisions by criteria determined at time points >four weeks apart may be too late to detect deterioration for timely retreatment [58]. It seems that biochemical markers are still not ideal for optimal retreatment decisions.

#### 5.2.2. Combined HBsAg/ALT Kinetics for Retreatment Decision

The quantitative HBsAg (qHBsAg) has been considered as a surrogate marker of transcriptionally active cccDNA or a marker of HBV-infected hepatocytes, and sensitive and reproducible qHBsAg assays have been available commercially and used widely in clinical practice and research [62]. Along with HBV DNA levels, serum HBsAg levels also upsurge prior to ALT elevation to its peak but may start to decline successively prior to the peak of ascending ALT in some hepatitis flares, which may resolve spontaneously [57]. Hepatitis flares with such a profile of combined HBsAg/ALT kinetics may reflect that the host immune response is dominating over the virus and the effective immune clearance of HBV is ongoing (host-dominating flare; HDF) (Figure 1a). In contrast, flares with qHBsAg increasing along with ascending ALT or remaining high after the peak of ALT may reflect that the virus is dominating over the host and the immune response is failing or being ineffective (virus-dominating flare; VDF) (Figure 1b) [58]. A proof-of-concept case study has shown that retreatment may be unnecessary or can be held in a patient with HDF, whereas timely retreatment was required for a patient with VDF [63]. This has been further confirmed by another study involving 48 patients who remained un-retreated, of whom five patients with off-NUC hepatitis flare have demonstrated off-NUC qHBsAg kinetics from EOT to HBsAg loss [64]. Furthermore, a retrospective appraisal study conducted by us using pre-retreatment combined HBsAg/ALT kinetics in 22 patients with off-NUC severe ALT flare (ALT > 30 × ULN), showing much greater and faster HBsAg decline (>1–3 log_10_ IU/mL decline in 12 months) during NUC retreatment in patients with VDF (bad flare but good response to NUC therapy), in contrast to minimal HBsAg decline or even HBsAg increase or rebound in patients with HDF (good flare but bad response to NUC therapy) [65]. An extension study using retrospective appraisal in 220 patients with off-NUC flare has further demonstrated that retreatment in patients with VDF achieved 1-year HBsAg decline of −0.91 log_10_ IU/mL in contrast to a decline of −0.07 log_10_ IU/mL in patients with HDF, while un-retreated patients with HDF showed greater HBsAg decline and a three-year HBsAg loss rate >20% (Jeng and Liaw, 2021, unpublished data). It is conceivable that patients with off-NUC VDF do need timely NUC retreatment to prevent severe ALT flare and help their ineffective immune response to fight the virus. In contrast, patients with off-NUC HDF which may overcome HBV and lead to further HBsAg decline and NUC retreatment should be withheld because it may halt or interrupt the strong endogenous immune clearance response of host. [66]. 

## 6. Conclusions and Perspective

Most real-world CHB patients preferred NUCs therapy. It is estimated that >90–95% of the patients were treated with NUCs. Most importantly, a paradigm shift from indefinite long-term or even lifelong to finite NUC therapy has been emerging globally. To identify the factors that can reliably predict the outcomes of CHB patients who stop NUCs therapy is quite important. Unfortunately, although there have been many studies trying to identify predictors of post-NUC events, accurate predictors have not been reliably identified [67]. So far, the best predictor is EOT qHBsAg <100 IU/mL, which showed a significantly lower relapse rate and higher probability of HBsAg loss after NUC discontinuation [48,68]. 

Proper monitoring is of paramount importance to ensure off-NUC safety. Patients with impending or overt hepatic decompensation definitely require immediate retreatment. Otherwise, “no retreatment” is an important determinant for off-NUC HBsAg loss. Hence, how to assess patients and decide whether to retreat or not are crucial in the strategy of finite NUC therapy. Of note is that patients with hepatitis flare with normal serum bilirubin and prothrombin time, a decision based on current biochemical markers, is not ideal, whereas combined HBsAg/ALT kinetics may help differentiate flares that need timely retreatment from those retreatment that can be held off or even deemed to be unnecessary. 

Finally, the development of new drugs that target different steps of the HBV life cycle, such as entry inhibitors, targeting cccDNAs, capsid inhibitors, or HBsAg excretion blockers, and focusing on improving host innate immunity such as lymphotoxin-B receptor agonist or Toll-like receptor agonist; or improving host adaptive immunity such as therapeutic vaccine or immune checkpoint inhibitors, is strongly anticipated. Most of the investigation agents are still in the preclinical or early phases of study. The strategy of finite NUC therapy has set a benchmark of high HBsAg loss rate. To be applicable, future drugs have to achieve an even higher HBsAg loss rate.

## Figures and Tables

**Figure 1 viruses-14-00434-f001:**
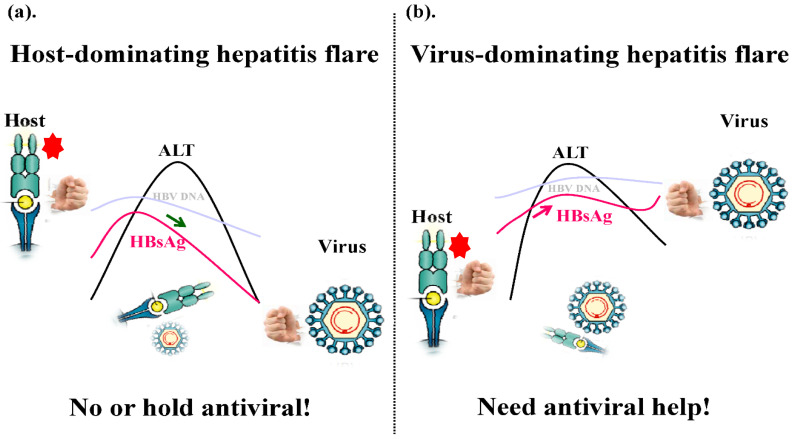
The cartoons demonstrated two types of combined qHBsAg/ALT kinetics during ALT flare in HBeAg-negative patients after end of NUC therapy. (**a**). “Host-dominating flare”: qHBsAg upsurged to peak levels and began to decline successively before or shortly after ALT peak. The flare was followed by a decline of HBV DNA and HBsAg levels with ALT normalization. (**b**). “Virus-dominating flare”: qHBsAg upsurged along with the ascending to the ALT to its peak and remained high after minor HBsAg decline. Patients may require antiviral therapy or hepatitis may persist or encounter another ALT flare later requiring antiviral(s) eventually. 
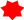
: representative of host antiviral immune power including cytolytic (HLA class-I restricted CD8 mediated hepatocytolysis) and non-cytolytic (IL-2; IFN-γ; TNF-α) antiviral immune ability.

**Table 1 viruses-14-00434-t001:** Comparison of HBsAg loss between finite and indefinite long-term NUC therapy.

Source [Reference]	(Country/Year)	No. of Patient	NUC Therapy	HBsAg Loss	Annular HBsAg Loss Rate
**Finite Therapy**					
Chan H.L., et al. [53]	(Hong Kong/2011)	53	LAM 3 Yr	23%/5 Yr	4.6%
Hadziyannis S.J., et al. [49]	(Greece/2012)	33	ADV 4–5 Yr	39%/5 Yr	7.8%
Chi H., et al. [54]	(Canada/2015)	59	NUC 5 Yr	14%/3 Yr	4.7%
Honer Zu Siederdissen C., et al. [50]	(Germany/2016)	15	NUC > 3 Yr	20%/4 Yr	5.0%
Berg T., et al. [51]	(Germany/2017)	21	TDF > 4 Yr	19%/3 Yr	6.3%
Papatheodoridis G.V., et al. [52]	(Greece/ 2018)	57	ETV/TDF 5 Yr	16%/1 Yr	16%
Jeng W.J., et al. [48]	(Taiwan/2018)	383 (CHB)	ETV/TDF 3 Yr	16%/6 Yr	2.7%
		308 (LC)		9%/6 Yr	1.5%
Chen C.H., et al. [55]	(Taiwan/2019)	234	ETV 3 Yr	13%/5 Yr	2.6%
**Indefinite long-term therapy**					
Chen C.H., et al. [55]	(Taiwan/2019)	226	ETV 7 Yr	1.8%/7 Yr	0.25%
Hsu Y.C., et al. [40]	(Multination/2021)	4769	ETV/TDF 5.2Yr	2%/10 Yr	0.22%

NUC: nucleos(t)ige analogue; LAM: Lamivudine; ADV: Adefovir; TDF: Tenofovir disoproxil fumarate; ETV: Entecavir; TAF: Tenofovir alafenamide; Yr: Years.

**Table 2 viruses-14-00434-t002:** HBsAg loss rate related to off-NUC events.

Event	HBV DNA (IU/mL)	ALT (U/L)	No of Patient	HBsAg Loss 6-Year Rate
Sustained response	<2000	N	144	36%
Virologic relapse	>2000	N	128	13%
Clinical relapse	>2000	>2 × ULN		
No-retreatment			150	19%
Re-treatment			269	1%
Total			691	13%

N: normal; ULN: upper limit of normal; data adapted from reference [48].

**Table 3 viruses-14-00434-t003:** Biochemical criteria for retreatment decisions.

Source [Reference]	Monitoring	Criteria to Retreat
Berg T., et al. [51]	2-weekly × 3 months 4-weekly~ *	ALT > 10 × ULN > 2 visit * 5–10 × ULN > 4 weeks * Bilirubin >1.5 mg/dL or INR >1.5
Papatheodoridis G.V., et al. [59]	Monthly × 3 months 3-monthly~	ALT > 10 × ULN # ALT > 5 × ULN + bilirubin > 2 mg ALT > 3 × ULN+ HBV DNA > 10^5^ IU/mL #
Liem K.S., et al. [60]	Week 4 and 6 then 6–8 weeks *	ALT > 15 × ULN # ALT >5 × ULN > 2 visits * ALT, 200–600 U/L for 6–8 weeks *
Garcia-Lopez M., et al. [61]	Week 3, 6, 12, 18, 24 *	ALT > 10 × ULN × 2 * ALT > 5–10 × ULN > 4 weeks * ALT > 2–5 × ULN > 6 months *

#: at one time point may be too early; *: follow-up >four-weeks may be too late.

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
