# Peer review of "Current Trend in Antiviral Therapy for Chronic Hepatitis B"

_viruses, 2022, doi:10.3390/v14020434_

Round 1

Reviewer 1 Report

The authors  summarised the different treatments used for HBV infection and their effects on viral and liver parameters. They also reported the literature data on the decision to retreatment.

The review is interesting and certainly of great clinical impact. However, some points need to be clarified and further investigated.

These are my main concerns:

  • Lines 81-86: The sentence “Peg-IFN has the... treatment [12].” is not very clear. Please to be rephrased.
  • Figure: The concept in the legend is not so clear and easy to understand by selling the figure shown. Please reformulate the figure by making it clearer.
  • It would be useful and very informative to create a table summarising the pros and cons (response, toxicity etc.) of the individual drugs used as well as their effect on virological and hepatic parameters (e.g. HBsAg, HBV-DNA, ALT etc.).
  • It is recommended to include a section on HBV treatment in case of co-infection (other hepatitis, HIV etc).
  • In line with the previous comment, it might be interesting to also define potential treatment approaches against HBV in pregnancy.
  • Correct Telbividine (Ldt) with Telbivudine (LdT).

Author Response

Dear editor and reviewer:

Thank you very much for your kind letter and constructive comments on our manuscript. My colleagues and I greatly appreciate the suggestions offered by you and the reviewers, and the opportunity to improve our manuscript. We have made the following changes point by point with changes underlined according to the suggestions of you and reviewers.

The reviewer 1 has 6 comments:

  1. Lines 81-86: The sentence “Peg-IFN has the... treatment [12].” is not very clear. Please to be rephrased.

Answer: Thank you very much for your kind suggestion. We have rephrased it and expressed the concept clearer (lines 84-88).

  1. Figure: The concept in the legend is not so clear and easy to understand by selling the figure shown. Please reformulate the figure by making it clearer.

Answer: Thank you very much for your valuable comment. We have added a star sign to further express the host antiviral immune ability and explain it in the figure legend. We hope it will make the figure concept clearer.

  1. It would be useful and very informative to create a table summarizing the pros and cons (response, toxicity etc.) of the individual drugs used as well as their effect on virological and hepatic parameters (e.g. HBsAg, HBV-DNA, ALT etc.).

Answer: Thank you very much for your kind suggestion. Even though 2 subcutaneous interferon (IFN)-based drugs and 7 oral nucleos(t)ide analogues (NUC) have been approved for the treatment of CHB. Lamivudine (LAM), adefovir (ADV) and telbivudine (LdT) have been almost obsoleted because of issues of drug resistance, whereas entecavir (ETV), tenofovir disoproxil fumarate (TDF), tenofovir alafenamide (TAF) and pegylated interferon (Peg-IFN) are the first-line drugs of choice for anti-HBV therapy. Recently, Roche pharmaceutics claimed no longer producing peg-IFN, only ETV, TDF and TAF left for clinical use. The latter three NUCs are quite similar in relevant issues including stronger antiviral power, high genetic barrier, and good safety profiles.

  1. It is recommended to include a section on HBV treatment in case of co-infection (other hepatitis, HIV etc).

Answer: Thank you very much for your kind suggestion. Because present review article tries to express current trend in antiviral therapy for chronic hepatitis B, especially the paradigm shift from indefinite treatment duration to finite therapy in HBeAg-negative CHB. The other relevant virus superinfection is not discussed in this issue. Our previous report of Taiwan consensus statement on chronic hepatitis B treatment has discussed this issue. (Please see reference 5).

  1. In line with the previous comment, it might be interesting to also define potential treatment approaches against HBV in pregnancy.

Answer: Answer: Thank you very much for your kind suggestion. Because present review article tries to express current trend in antiviral therapy for chronic hepatitis B, especially the paradigm shift from indefinite treatment duration to finite therapy in HBeAg-negative CHB. The relevant issue of HBV and pregnancy is not discussed here. Our previous report of Taiwan consensus statement on chronic hepatitis B treatment has discussed HBV and pregnancy issue. (Please see reference 5).

  1. Correct Telbividine (Ldt) with Telbivudine (LdT).

Answer: Thank you so much for your kind reminding. We have corrected it accordingly. (Lines 79, 180)

Reviewer 2 Report

The aim of this manuscript is to discuss the forefront and evolving trend in the antiviral therapy of chronic HBV infection. In this review, the authors convincingly evaluate that current and most applied antiviral therapy, as a valid strategy for chronic infection with Hepatitis B virus.

Even if the manuscript provides an organic overview, with a densely organized structure and based on well-synthetized evidence, there are aspects to be mentioned, to make the article fully readable. For these reasons, the manuscript requires minor changes.

Please find below an enumerated list of comments on my review of the manuscript:

INTRODUCTION:

LINE 34, 41 and 52: Hepatitis B

LINE 34: Hepatitis B Virus is the major cause of acute and chronic liver disease and a global health problem, with a significant impact on everyday life. This issue is also highlighting by several and recent studies (see, for reference: Mastrodomenico, M.; Muselli, M.; Provvidenti, L.; Scatigna, M.; Bianchi, S.; Fabiani, L. Long-term immune protection against HBV: Associated factors and determinants. Hum. Vaccines Immunother. 2021), which also investigate the importance of a persistent and long-term immunogenicity of HBV vaccine.

LINE 101: Entecavir, a cyclopentyl guanosine analogue, is considered a first – line treatment for chronic Hepatitis B, due to its high antiviral efficacy and low rate resistance (see, for reference: Choi, W. M., Choi, J., & Lim, Y. S. (2021). Effects of tenofovir vs entecavir on risk of hepatocellular carcinoma in patients with chronic HBV infection: a systematic review and meta-analysis. Clinical Gastroenterology and Hepatology19(2), 246-258). In this section, the manuscript may beenfit from introducing recent reference, as previously mentioned.

In conclusion, this manuscript is densely presented and well organized, based on well-synthetized evidences. The authors were lucid in their style of writing, making it easy to read and understand the message, portrayed in the manuscript. Besides, the methodology design was rigorous and appropriately implemented within the study. However, many of the topics are very concisely covered. This manuscript provided a comprehensive review of current knowledge in this field. Moreover, this research have futuristic importance and could be potential for future research. However, I have minor comments only for the introductive section, for improvement before acceptance for publication. The article is accurate and provides relevant information on the topic and I suggest minor changes to be made in order to maximize its scientific impact. I would accept this manuscript, if the comments are addressed properly.

Author Response

Dear editor and reviewer:

Thank you very much for your kind letter and constructive comments on our manuscript. My colleagues and I greatly appreciate the suggestions offered by you and the reviewers, and the opportunity to improve our manuscript. We have made the following changes point by point with changes underlined according to the suggestions of you and reviewers.

The reviewer 2 has 3 comments:

  1. LINE 34, 41 and 52: Hepatitis B

Answer: Thank you very much for your kind reminding. We have corrected them accordingly. (Lines 34, 43, 55)

  1. LINE 34: Hepatitis B Virus is the major cause of acute and chronic liver disease and a global health problem, with a significant impact on everyday life. This issue is also highlighting by several and recent studies (see, for reference: Mastrodomenico, M.; Muselli, M.; Provvidenti, L.; Scatigna, M.; Bianchi, S.; Fabiani, L. Long-term immune protection against HBV: Associated factors and determinants. Hum. Vaccines Immunother. 2021), which also investigate the importance of a persistent and long-term immunogenicity of HBV vaccine.

Answer: Thank you very much for your valuable comments. We have added the primary prevention of HCC by statement of universal vaccination program and recruit your suggesting reference as reference 4 in our manuscript. (Lines 41-42 and reference 4)

  1. LINE 101: Entecavir, a cyclopentyl guanosine analogue, is considered a first – line treatment for chronic Hepatitis B, due to its high antiviral efficacy and low rate resistance (see, for reference: Choi, W. M., Choi, J., & Lim, Y. S. (2021). Effects of tenofovir vs entecavir on risk of hepatocellular carcinoma in patients with chronic HBV infection: a systematic review and meta-analysis. Clinical Gastroenterology and Hepatology19(2), 246-258). In this section, the manuscript may beenfit from introducing recent reference, as previously mentioned.

Answer: Thank you again for your valuable suggestion. We also added this statement in the introduction section but recruit another paper for reference because the paper contains for relevant researches for systematic analysis. ( Lines 45-46 and reference 6)
